# The Photic Stimulation Has an Impact on the Reproduction of 10 s Intervals Only in Healthy Controls but Not in Patients with Schizophrenia: The EEG Study

**DOI:** 10.3390/brainsci13010112

**Published:** 2023-01-07

**Authors:** Galina V. Portnova, Aleksandra V. Maslennikova

**Affiliations:** 1Institute of Higher Nervous Activity and Neurophysiology of the Russian Academy of Sciences, Butlerova 5A, Moscow 117485, Russia; 2Pushkin State Russian Language Institute, Akademika Volgina st, 6, Moscow 117485, Russia

**Keywords:** EEG, time reproduction, schizophrenia, time perception, envelope mean frequency, Hjorth complexity

## Abstract

Schizophrenia is a mental disorder characterized by both abnormal time perception and atypical relationships with external factors. Here we compare the influence of external photic stimulation on time production between healthy subjects (*n* = 24) and patients with schizophrenia (*n* = 22). To delve into neuropsychological mechanisms of such a relationship, the EEG was recorded during variable conditions: during production of 10 s intervals; during photic stimulation of 4, 9, 16, and 25 Hz; and during combinations of these conditions. We found that the higher frequency of photic stimulation influenced the production of time intervals in healthy volunteers, which became significantly longer and were accompanied by corresponding EEG changes. The impact of photic stimulation was absent in patients with schizophrenia. In addition, the time production was characterized by less accuracy and the absence of EEG dynamics typical for healthy controls that included an increase in alpha2 power and envelope frequency. Our findings indicated that the time perception was not adjusted by external factors in patients with schizophrenia and might have involved cognitive and mental processes different from those of healthy volunteers.

## 1. Introduction

The subjective time perception is dependent on many environmental and internal factors, which contribute not only to the person’s subjective experience of the passage of time but also to the accuracy of the assessment and reproduction of time units [1]. In particular, people who experience stressful or frightening events remember that time seemed to slow down at that moment [2] or, oppositely, estimate subjective speeding up of time, when they fall asleep [3]. The subjective flow or passage of time could be also related to the use of various psychoactive drugs, such as caffeine, dopamine, marijuana, ethanol, etc. [4,5,6].

The sense of time and the ability to estimate time intervals also depends on mental or neurological pathology. In particular, patients with schizophrenia demonstrate the acceleration of the “internal clock”, which correlates with the severity of positive symptoms [7], and a distorted sense of time, which results in positive and negative symptoms [8]. Other studies have demonstrated that patients with schizophrenia have low accuracy of time perception but high variability of internal clock, which could be associated with medication influences [9]. Finally, some researchers have doubted the primacy dysfunction of time perception and suggested that abnormalities in time perception in this group of patients could be secondarily associated with impaired cognitive processes [10].

According to the recent models of time perception, the subjective sense of time is associated with the number of rhythmic repetitive external or internal events that are implicitly perceived by a person as units of time. The mentioned internal events could be associated with heart or breathing rate [11,12,13], and external events include the clock speed [14,15], flicker frequency, spinning speed [16], and frequency of photic stimulation [17]. Despite the lack of consensus on the mechanisms of timing deficit in patients with schizophrenia, this group of patients is also interesting because of the independence of their cognitive processes from external factors. It is well known that patients with schizophrenia demonstrate nonconforming behaviors [18,19], which usually contribute to the impairment in functioning of orbitofrontal cortex and hippocampus and executive decision-making processes [20,21]. At the same time, the influence of the environmental factors on time perception and other cognitive functions in patients with schizophrenia remains poorly understood.

The main research hypothesis was the difference in the impact of rhythmic stimulation of variable frequencies (namely, photic stimulation) on the time perception between healthy volunteers and patients with schizophrenia. The photic stimulation was selected as an environmental factor as being a routine method in clinical electroencephalography with a well-known potential effect on EEG dynamics. In particular, the photic stimulation could provoke epileptiform discharges, could be a trigger for patients with photosensitive epilepsy and migraine, and could induce other abnormalities in the patient’s EEG [22,23,24]. At the same time, in healthy adults, photic stimulation did not provoke abnormalities, but the flashing lights at certain frequencies induced the appearance of this rhythm’s harmonics in brain oscillations [25] or sub-harmonics (photic driving response) or induced quantitative changes of EEG that were inaccessible for visual analysis, which depended on age and cognitive activity of subjects [17]. Finally, the previous findings demonstrated that the reactive changes of the EEG to the external stimuli could be effectively assessed by the non-linear features of EEG, especially in the cases of patients with mental or neurological pathology [26,27]. Moreover, some of the non-linear features, for example, Hjorth complexity or fractal dimension, were sensitive to specific perceptual processes which could contribute to the psychiatric and neurological pathology [28,29,30].

## 2. Methods

### 2.1. Participants

Healthy volunteers were recruited through online or institution advertisements. In total, 24 participants of the control group (mean = 26.7 years, SD = 5.1, 13 female, nine male) with no history of schizophrenia or schizoaffective disorder or medical illness associated with increased rates of depression completed the study. All patients participating in the study did not have pathologies of the auditory system. Participants gave informed consent for their participation. The work was approved by the ethics committee of the Institute of Higher Nervous Activity and Neurophysiology of the Russian Academy of Sciences (protocol no. 30 April 2021).

Twenty-two patients with schizophrenia (mean = 26.4 years, SD = 5.2, 10 female, 12 male) were assessed with a diagnostic clinical interview (ICD-10: F2*) to determine the diagnosis. Individuals with a history of other mental or neurological diagnoses were also included. The severity of the symptoms was assessed using the PANSS scale (mean was 102.17 ± 15.7). All patients were recruited from Psychiatric Clinical Hospital, named after N. A. Alekseev, due to psychotic episodes within 7–14 days after hospitalization and received typical treatment with antipsychotic medications. All patients received haloperidol 15.9 ± 3.2 mg/day and aminazine 137.5 ± 42 mg/day.

All subjects were right-handed and had normal hearing level in both ears, and their intellectual skills were within the normal range. They had no history of neuropsychiatric disorders or head injury.

### 2.2. Stimuli and Paradigm of the Study

Our study consisted of four series: (1) two-minute resting state EEG with eyes closed; (2) the production of 10 s intervals (we asked participant to do it three times); (3) photic stimulation of 4, 9, 16, and 25 Hz, which was presented for 30 s each in random order; and (4) photic stimulation of 4, 9, 16, and 25 Hz during the reproduction of 10 s intervals (twice for each condition).

The photic stimulation was presented using the standard technique for intermittent photic stimulation as a series of photic flashes delivered by a strobe light at a distance of 20–30 cm from the participant’s face with the eyes closed.

### 2.3. The Task of Time Production

To estimate time, we asked participants to press the spacebar button when they started to estimate time and then press it again when the 10 s interval was over.

After the study was completed, we asked participants to describe how they estimated the time. A total of 24 of the initial 25 subjects of the control group and 21 of the 25 patients reported that they used inner speech numbering for the task of time production. One healthy participant reported that he used his sense of time without numbering, one patient reported that he could “see the time”, and two patients could not remember how they estimated the time. The latter participants were excluded from the study.

### 2.4. Analysis of Time Perception

We analyzed three parameters of the time production:Real time (Treal).

Participants varied their produced time intervals from 5692 to 13,989 ms in healthy participants and from 4334 to 18,662 ms in patients with schizophrenia, so we added two other parameters (see Table 1):2.Normalized time (Tnorm), i.e., the original values of time intervals were normalized for each participant using mean and std Treal.3.The error of time perception (Terr): $\sqrt{(10,000-{\mathrm{Treal}}^{2}}$).

### 2.5. EEG Registration

The EEG was acquired using a 19-channel EEG amplifier Encephalan with the recording of polygraphic channels (Poly4, Medicom MTD, Taganrog, Russia) during 10 min. The sampling rate was 250 Hz. The amplifier band-pass filter was nominally set to 0.05–70 Hz. AgCl electrodes (Fp_1_, Fp_2_, F_7_, F_3_, Fz, F_4_, F_8_, T_3_, C_3_, Cz, C_4_, T_4_, T_5_, P_3_, Pz, P_4_, T_6_, O_1_, and O_2_) were placed according to the International 10–20 system. The electrodes placed on the left and right mastoids served as joint references under unipolar montage. The vertical EOG was recorded with AgCl cup electrodes placed 1 cm above and below the left eye, and the horizontal EOG was acquired by electrodes placed 1 cm lateral from the outer canthi of both eyes. The electrode impedances were kept below 10 kΩ.

### 2.6. EEG Preprocessing

Continuous EEG corresponding to stimulation and the resting state of each subject was cleaned from eye movements by an ICA-based algorithm in the EEGLAB plugin for MATLAB 7.11.0 (Mathworks Inc., Natick, MA, USA) and the ALICE toolbox [31]. Muscle artifacts were cut out through manual data inspection. The continuous resting-state EEG of each subject was filtered with a band-pass filter at 0.5–30 Hz.

### 2.7. Data Analysis

We analyzed data over three regions: frontal (F3, Fz, and F4), central (C3, Cz, and C4), and parietal (P3, Pz, and P4).

#### 2.7.1. Power Spectral Density (PSD)

Fast Fourier transform was used to analyze power spectral density (PSD). The EEG spectrum was estimated for every 178 ± 22.3 s interval. The resulting normalized spectra were integrated over intervals of unit width in the range of interest (2–3 Hz, 3–4 Hz,…29–30 Hz).

We analyzed the PSD for all experimental conditions in the following bands: 2–8 Hz (slow-wave band), 8–10 Hz (alpha1 band), 10–13 Hz (alpha2 band), and the beta band, which we divided into 14–17 Hz and 17–20 Hz sub-diapasons.

#### 2.7.2. Fractal Dimension (FD)

We performed the calculations of the examined signal band-pass-filtered in the range of interest (1.6–30 Hz); a Butterworth 12th-order filter was used. Fractal dimension (FD) was evaluated using the Higuchi algorithm.

#### 2.7.3. Envelope Mean Frequency (EMF)

To evaluate the (de-)synchronization dynamics of the rhythms, we applied the following method. First, we calculated the envelope of the EEG signal for the whole frequency range (1.6–30 Hz) using the Hilbert transform.

#### 2.7.4. Hjorth Complexity

Hjorth complexity (HC) represents the change in frequency and indicates how the shape of a signal is similar to a pure sine wave. This parameter was calculated for the wideband 1.6–30 Hz filtered signal in the following way: complexity(y(t)) = mobility(dy(t)/dt)/mobility(y(t)), where mobility(y(t)) = √ (var(dy(t)/dt/var(y(t))), y(t) is a signal, y’(t) is its derivative, and var(...) is the variance.

#### 2.7.5. Statistical Analysis

Factorial and repeated measures ANOVAs with the following post hoc comparison (Bonferroni, *p* < 0.05) were used to determine group effects on the EEG dynamics and the results of time reproduction (both behavioral and EEG). Only significant effects supported by the post hoc Bonferroni test were discussed further; the figures presented in the manuscript represent only the method results tested by the Bonferroni correction.

We analyzed a possible association of the EEG metrics with the age, sex, PANSS scores, dose of applied medicine, and duration of measured time intervals using the Spearman correlation analysis corrected for multiple comparisons by the cluster-based permutation test using the clustering method (MATLAB toolbox for BCI) with 500 permutations at each node (the Bonferroni corrected *p*-value of 0.05). For the only significant correlations which were found after the permutation test, we applied additional correlation analysis between testing scores, behavioral answers, and the EEG dynamics over three selected regions; we used Spearman’s rank correlation coefficient and adjusted *p*-values. Further, the adjusted *p*-values of the significant correlations are presented in the manuscript.

## 3. Results

### 3.1. The Time Perception

The produced time (Treal) did not differ significantly between groups. The Tnorm increased significantly during photic stimulation only in healthy subjects (F(4, 180) = 3.8359, *p* = 0.00512, partial eta-squared = 0.14): the Tnorm for the 9, 16, and 25 Hz photic stimulation was significantly higher compared with the series with a single time reproduction and 4 Hz photic stimulation, and also, the Tnorm for the 16 and 25 Hz photic stimulation was significantly higher compared with the 9 Hz photic stimulation (post hoc Bonferroni test, *p* < 0.05; see Figure 1). The error of time production was significantly higher in the group of patients compared with the control group (F(1, 45) = 5.6791, *p* = 0.02145, partial eta-squared = 0.11).

### 3.2. The Power Spectral Density

We did not find any significant group difference between the PSDs of the analyzed bands during the resting state. During time production, the PSD of the alpha2 rhythm was significantly higher in healthy participants compared with the group of patients in the frontal and central areas (F(1, 45) = 7.2503, *p* = 0.00356, partial eta-squared = 0.16).

We found that during the 16 Hz photic stimulation, the PSD of 14–17 Hz was significantly higher over the central, frontal, and parietal areas in the control group compared with the group of patients F(1, 45) = 5.6971, *p* = 0.01256, partial eta-squared = 0.11). Other changes in the PSD were registered in both groups of participants: significant increases in the 8–10 Hz and 17–20 Hz PSDs over the central, frontal, and parietal areas during 9 Hz photic stimulation (F(1, 45) = 5.7765, *p* = 0.01654, partial eta-squared = 0.10).

During the production of 10 s intervals synchronized with photic stimulation, patients did not demonstrate significant differences between experimental conditions. However, in healthy subjects, time reproduction accompanied by 9 Hz photic stimulation induced significant increases in the 8–10 Hz and 17–20 Hz PSDs over the central, frontal, and parietal areas; 16 Hz stimulation caused an increase in the 14–17 Hz PSD; and 25 Hz caused an increase in the 10–13 Hz PSD (F(51, 1836) = 2.9707, *p* = 0.00000, partial eta-squared = 0.19).

Thus, in spite of photic stimulation inducing similar PSD changes, although significantly reduced compared with the control subjects, the additional task of time production in patients with schizophrenia led to the complete disappearance of PSD dynamics associated with photic stimulation, whereas the PSD changes in healthy subjects during photic stimulation accompanied by the time reproduction task became even more pronounced (Figure 2).

### 3.3. Non-Linear Features of EEG

The fractal dimension was significantly higher in the group of patients independently of the experimental conditions (F(1, 38) = 6.7726, *p* = 0.0088, partial eta-squared = 0.15). The envelope mean frequency (EMF) was also significantly higher in patients with schizophrenia, however, only during time production. In particular, the EMF did not differ between patients and healthy subjects during the resting state; however, it significantly increased compared with the resting state in patients with schizophrenia during the production of time intervals and oppositely decreased in healthy volunteers (F(1, 45) = 8.7533, *p* = 0.00506, partial eta-squared = 0.17) (see Figure 3A).

The Hjorth complexity increased significantly in healthy subjects during photic stimulation; this effect was not found in the group of patients (F(9, 306) = 2.5271, *p* = 0.00830, partial eta-squared = 0.09). The group difference between Hjorth complexities was found for photic stimulation from 4 to 16 Hz (F(1, 38) = 5.9496, *p* = 0.01759, partial eta-squared = 0.139): it was significantly higher in the subjects of the control group (see Figure 3B).

### 3.4. The Correlation Analysis

We did not find significant correlations between age, sex, the PANSS scores, the dose of applied medicine, and the results of time perception or EEG dynamics. The significant results of the correlation analysis between time production and the corresponding EEG changes are presented below.

The PSD of the alpha2 rhythm (10–13 Hz) of healthy participants when they were asked to produce 10 s intervals during 25 Hz photic stimulation significantly correlated with Tnorm for this experimental task (r = 0.53, 0.0055).

The correlation analysis showed that the higher Hjorth complexity in healthy participants during time production accompanied by photic stimulation (4, 9, and 16 Hz) correlated with higher Tnorm (r = 0.54, 0.005; r = 0.49, 0.011; r = 0.44, 0.027). In patients with schizophrenia, the Hjorth complexities were oppositely negatively correlated with time production (Treal) during 4, 9, and 16 Hz photic stimulation (r = −0.59, 0.0039; r = 0.49, 0.019; r = 0.52, 0.013) (see Figure 4).

## 4. Discussion

According to our results for time production, we replicated the previous findings demonstrating that patients with schizophrenia were less accurate during time production [7,32]. In addition to the fact that the mechanisms of the time deficits in patients with schizophrenia were not completely investigated, most studies concluded that their low time perception accuracy could be explained by the impairment of cognitive processes, particularly working memory and attention [7,10], and by antipsychotic medications or abnormalities in dopamine transmissions [33,34]. Our EEG findings also supported the hypothesis of disabled cognitive processing during time perception in a group of patients. In particular, during time production (without additional photic stimulation), healthy participants unlike patients with schizophrenia demonstrated significant increases in alpha2 rhythm PSD, and enhanced envelope frequency contributed to higher mental, cognitive, and emotional activity [35,36,37].

The main findings of the study were that only the healthy participants demonstrated a significant increase in normalized time production during simultaneous 9, 16, and 25 Hz photic stimulation. For the patients with schizophrenia, time production was independent of the photic stimulation rate. The EEG dynamics corresponded to the time production during photic stimulation, and the results of the correlation analysis also supported that the frequency of photic stimulation influenced only the time reproduction of healthy volunteers. In particular, the increase in the duration of the 10 s interval (Tnorm) in healthy volunteers correlated with the power of the alpha2 rhythm. The increase in the 10–13 Hz PSD was attributed only to the time production accompanied by 25 Hz photic stimulation. Further, the increase in Hjorth complexity in healthy participants was attributed to the experimental block with simultaneous production of time interval and photic stimulation (4, 9, and 16 Hz) correlated with higher Tnorm. So, the significant increase in Hjorth complexity and alpha2 rhythm PSD in healthy participants contributed to the higher increase in Tnorm. The effect was absent in patients with schizophrenia, who had no increase in Hjorth complexity and alpha2 rhythm PSD during photic stimulation and time perception, did not change the duration of the time interval, and moreover showed a negative correlation between time production and Hjorth complexity. The higher alpha rhythm amplitude was previously associated with higher attention and successful recognition during cognitive tasks [38]. Other studies reported that creative cognition usually induced the elicited synchronization in the upper alpha band [39]. So, the association with the large duration of Tnorm during the 25 Hz photic increase in the 10–13 Hz PSD could be a sign of specific cognitive activity involved in the time reproduction of healthy volunteers. In this case, the subjects demonstrated higher workloads and were more likely to be task involved, so the perceived time decreased with the increasing mental workload [40,41].

The higher produced duration during 9 and 16 Hz photic stimulation presumably has slightly different mechanisms, which are also supported by the EEG dynamics. In particular, the enhanced Hjorth complexity in healthy participants was previously attributed to the sensory processing [42]. Therefore, the dynamic of the perceived time duration as an impact of photic stimulation could be considered as an “adjustment” or “adaptation” of individuals’ perceptual systems in accordance with the parameters of the external events. Different studies previously demonstrated the impact of rhythmic environmental stimuli on the sensory systems when they revealed the effects of brain plasticity during musical rhythm training [43,44] or rhythmical sweeping [45]. We assumed that the absence of the adjustment in patients with schizophrenia could be a sign of cognitive rigidity and the deficit of plasticity in sensory systems during any mental activity [46]. The discussed assumptions were also consistent with the previous EEG findings demonstrating controversial changes of the Hjorth complexity between healthy adults and patients with severe neurological impairment during listening to emotional sounds [26] and indicated severe impairments in sensory information processing.

### Limitations

The number of participants was relatively small, and the 2:1 ratio (healthy participants to patients) was not provided. To avoid the sample size limitation, we attempted to equalize the target and control groups by their age and sex.

## 5. Conclusions

Our findings indicated that the impact of the photic stimulation on the time production and EEG changes was absent in patients with schizophrenia. In addition, the time production was characterized by less accuracy and the absence of EEG dynamics typical for healthy controls that included an increase in alpha2 power and envelope frequency. We suggested that impairments in sensory information processing including variable cognitive and mental processes resulting in the lack of EEG and behavioral changes corresponded to the impact of the external factors during the time perception.

## Figures and Tables

**Figure 1 brainsci-13-00112-f001:**
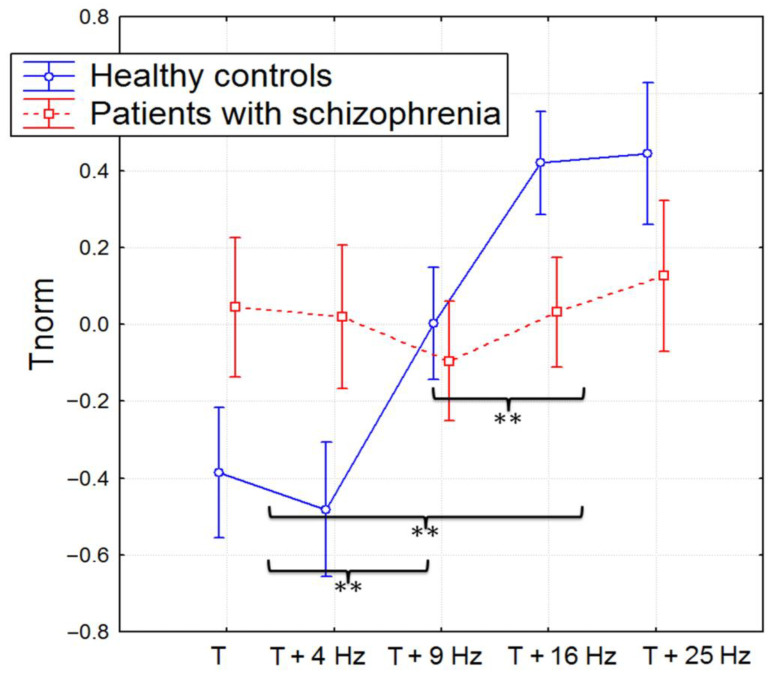
The mean values of the normalized time (Tnorm: mean ± SD) during different experimental blocks (time production and time production accompanied by photic stimulation (4, 9, 16, and 25 Hz)) in healthy participants and patients with schizophrenia. **—*p* <0.01.

**Figure 2 brainsci-13-00112-f002:**
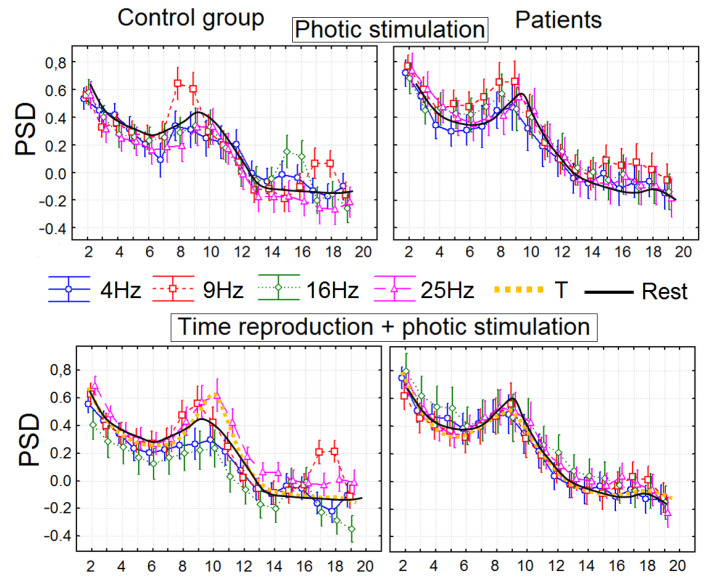
Power spectral density (PSD) in two groups of subjects depicted in the 2–20 Hz diapason with 1 Hz steps for 10 experimental blocks: block 1, resting state (black line in upper and bottom images); block 2, time reproduction (yellow line in bottom images); blocks 3–6, the 4, 9, 16, and 25 Hz photic stimulation (blue, red, green, and pink lines in upper images); and blocks 7–10, reproductions of time intervals simultaneously with photic stimulation (blue, red, green, and pink lines in bottom images).

**Figure 3 brainsci-13-00112-f003:**
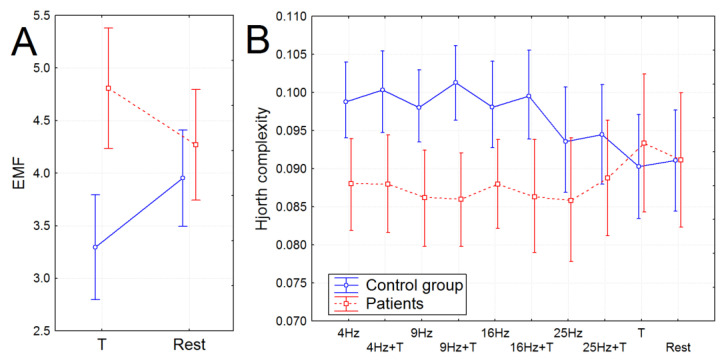
The non-linear features of EEG in healthy subjects and patients with schizophrenia. (**A**) EMF during time production (T) and the resting state (Rest). (**B**) Hjorth complexity during photic stimulation (4 Hz, 9 Hz, 16 Hz, and 25 Hz) and time reproduction accompanied by photic stimulation (4 Hz + T, 9 Hz + T, 16 Hz + T, and 25 Hz + T). **—*p* <0.01.

**Figure 4 brainsci-13-00112-f004:**
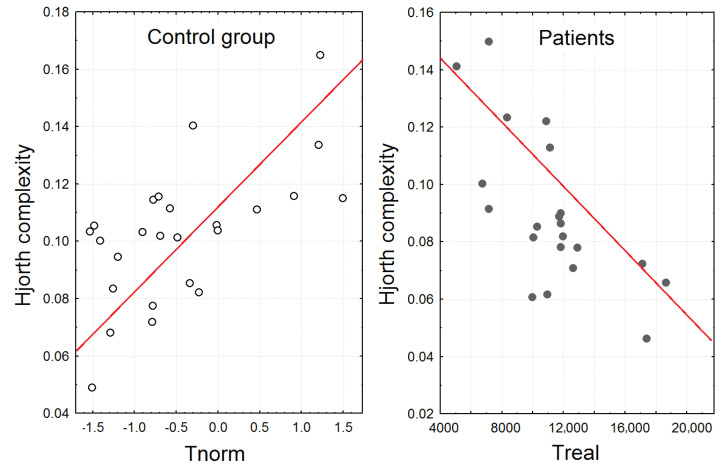
Scatterplots demonstrating the association between Hjorth complexity (averaged over the central and frontal regions) during the time reproduction task accompanied by 9 Hz photic stimulation and the duration of time reproduction during this experimental block (Tnorm for healthy participants and Treal for patients).

**Table 1 brainsci-13-00112-t001:** The table of descriptive statistics. T—time production; 4–25 Hz—time production during photic stimulation.

Time Parameters		Valid *N*	Mean	S.D.	Valid *N*	Mean	S.D.
	PANSS	22	102.17	15.69	-		
Treal	T	22	11,279	2689	24	9677	1939
	4 Hz	22	11,841	2971	24	9765	2344
	9 Hz	22	10,836	2569	24	10,068	2181
	16 Hz	22	11,155	2346	24	10,321	1821
	25 Hz	22	11,539	2475	24	10,231	2149
Tnorm	T	22	0.05	0.81	24	−0.39	0.88
	4 Hz	22	0.02	0.91	24	−0.48	0.84
	9 Hz	22	0.00	0.64	24	0.00	0.80
	16 Hz	22	0.03	0.67	24	0.42	0.67
	25 Hz	22	0.13	1.02	24	0.44	0.82
Terr	T	22	294.76	256.18	24	158.79	115.88
	4 Hz	22	358.10	390.80	24	198.63	126.83
	9 Hz	22	268.05	250.11	24	176.21	128.68
	16 Hz	22	286.32	208.14	24	178.83	152.75
	25 Hz	22	333.00	336.31	24	164.38	140.49

## Data Availability

The data could be available on the request.

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
