# Peer review of "The Photic Stimulation Has an Impact on the Reproduction of 10 s Intervals Only in Healthy Controls but Not in Patients with Schizophrenia: The EEG Study"

_brainsci, 2023, doi:10.3390/brainsci13010112_

Round 1

Reviewer 1 Report

I really appreciate the opportunity to review the manuscript  entitled:
"The photic stimulation has an impact on the reproduction of 10-s intervals only in healthy controls, but not in patients with schizophrenia: the EEG study"

I commend the authors for describing this critical and timely issue. The paper is interesting and well-written; however, I would like to highlight an issue that merits revision:

The control group is not large enough, to have good support in a case-control study it would be good to have a 2:1 ratio of controls to cases (2 controls for each case), please authors to include this among the data limitations.

Author Response

Reviewer 1

We would like to thank the reviewer for a reading our revised manuscript and appreciate the reviewer’s comments. Following the reviewer suggestions we corrected the manuscript.

Please find our responses to your comments below:

The control group is not large enough, to have good support in a case-control study it would be good to have a 2:1 ratio of controls to cases (2 controls for each case), please authors to include this among the data limitations.

We apologize for the reduced number of control subjects. We attempted to compensate the ~1:1 ratio by the careful selection of subjects of the control group by age and sex.  We added the limitation section to the manuscript.

Reviewer 2 Report

The manuscript of Galina V. Portnova and Aleksandra V. Maslennikova entitled „The photic stimulation has an impact on the reproduction of 10-s intervals only in healthy controls, but not in patients with schizophrenia: the EEG study.” reveals interesting results about schizophrenia. The manuscript is generally well-written, although I would have some minor comments to improve the manuscript:

 1. It should be mentioned in 2.1, that mean = 26.7 and 26.4 means age (probably).

2. How is it possible, that all of the patients with schizophrenia included in the study received the same combination of medication (Haloperidol 15.9±3.2 85 mg /day and Aminazine 137.,5±42 mg /day)? Were the patients selected based on this or is it a coincidence?

3. Table 1 should be moved to the end of 2.4, where the terms are defined or maybe to the results section (+ citing the table in the text in the same place). Decimals should be indicated in the table also wit dot (.) instead of comma (,).

Author Response

We appreciate the reviewer’s insightful comments and thank the reviewer for very thorough reading our manuscript. We tried to take all of them into account, what, we believed, resulted in the considerable improvement of the article. Following the reviewer suggestions and comments we corrected the text. The more detailed responses to the reviewer’s suggestions and comments are the following:

  1. It should be mentioned in 2.1, that mean = 26.7 and 26.4 means age (probably).

Thank you for the comment. It’s an age. We corrected the manuscript

  1. How is it possible, that all of the patients with schizophrenia included in the study received the same combination of medication (Haloperidol 15.9±3.2 85 mg /day and Aminazine 137.,5±42 mg /day)? Were the patients selected based on this or is it a coincidence?

We initially recruited the patients according to their symptoms and diagnoses and excluded from the analysis patients with untypical combination of medications. At the same time, the combination of Haloperidol and Aminazine is a typical combination of medication for our psychiatric hospitals and usually applied for all patients with schizophrenia in acute psychiatric crisis if they didn’t continuously applied other medications. So, initially we recruited  24 patients and two of them had applied previously the Rispolept prospect with a combination of several antidepressants and Atarax (or Adaptol) and psychiatrists only adjusted the dosage of drugs, but did not replace them.  

  1. Table 1 should be moved to the end of 2.4, where the terms are defined or maybe to the results section (+ citing the table in the text in the same place). Decimals should be indicated in the table also wit dot (.) instead of comma (,).

We corrected the text following the reviewer suggestions